# Single-cell and bulk RNA sequencing analysis reveals CENPA as a potential biomarker and therapeutic target in cancers

Hengrui Liu[1,2], Miray Karsidag[3], Kunwer Chhatwal[4], Panpan Wang[5]*, Tao Tang[6]*

1 Cancer Research Institute, Jinan University, Guangzhou, Guangdong, China, 2 Yinuo Biomedical Co., Ltd, Tianjin, China, 3 Canyon Crest Academy, San Diego, CA, United States of America, 4 Hopkinton High School, Hopkinton, MA, United States of America, 5 The First Affiliated Hospital of Jinan University, Guangzhou, Guangdong, China, 6 Sun Yat-Sen University Cancer Center, Guangzhou, Guangdong, China

* tangtao@sysucc.org.cn (TT); wangpp@jnu.edu.cn (PW)

## Abstract

### Background

Cancer remains one of the most significant public health challenges worldwide. A widely recognized hallmark of cancer is the ability to sustain proliferative signaling, which is closely tied to various cell cycle processes. Centromere Protein A (CENPA), a variant of the standard histone H3, is crucial for selective chromosome segregation during the cell cycle. Despite its importance, a comprehensive pan-cancer bioinformatic analysis of CENPA has not yet been conducted.

### Methods

Data on genomes, transcriptomes, and clinical information were retrieved from publicly accessible databases. We analyzed CENPA's genetic alterations, mRNA expression, functional enrichment, association with stemness, mutations, expression across cell populations and cellular locations, link to the cell cycle, impact on survival, and its relationship with the immune microenvironment. Additionally, a prognostic model for glioma patients was developed to demonstrate CENPA's potential as a biomarker. Furthermore, drugs targeting CENPA in cancer cells were identified and predicted using drug sensitivity correlations and protein-ligand docking.

### Results

CENPA exhibited low levels of gene mutation across various cancers. It was found to be overexpressed in nearly all cancer types analyzed in TCGA, relative to normal controls, and was predominantly located in the nucleus of malignant cells. CENPA showed a strong association with the cancer cell cycle, particularly as a biomarker for the G2 phase. It also emerged as a valuable diagnostic and prognostic biomarker across multiple cancer types. In glioma, CENPA demonstrated reliable prognostic potential when used alongside other prognostic factors. Additionally, CENPA was linked to the immune microenvironment.

**Data Availability Statement:** The data underlying the results presented in the study are available from open database.

**Funding:** This study was supported by funding from the K. C. Wong Education Foundation, the

Natural Science Foundation of China (81603342) awarded to PW, the Guangdong Basic and Applied Basic Research Foundation (2022A151501264, 2022A1515012641, 2024A1515012948) awarded to PW, the Guangzhou Science and Technology Project (SL2023A03J00309, 2024A03J0154, 2023B01J1004) awarded to PW, and the Guangdong Provincial Bureau of Traditional Chinese Medicine Research Project (20221107) awarded to PW.

**Competing interests:** The authors have declared that no competing interests exist.

Drugs such as CD-437, 3-Cl-AHPC, Trametinib, BI-2536, and GSK461364 were predicted to target CENPA in cancer cells.

## Conclusion

CENPA serves as a crucial biomarker for the cell cycle in cancers, offering both diagnostic and prognostic value.

## Introduction

Cancer is one of the most concerning public health issues in the world [1, 2]. Many common molecular pathological mechanisms shared across different neoplastic diseases have been identified to facilitate clinical cancer diagnosis, prognosis, and therapies. Cancer databases, such as The Cancer Genome Atlas (TCGA) [3], Genotype-Tissue Expression (GTEx) [4], the Chinese Glioma Genome Atlas (CGGA) [5], and the International Cancer Genome Consortium (ICGC) [6], provide gene alteration, gene expression, and clinical information on different cancer types, facilitating pan-cancer studies for identification and understanding of targets or biomarkers that exert common effects across cancer types. Although there is biases and limitations [7], these databases have been wildly used in many previous studies [8–29].

Six hallmarks of cancer have been proposed to constitute an organizing principle that provides a logical framework for understanding the remarkable diversity of cancers [30]. One of the cancer hallmarks wildly accepted is sustaining proliferative signaling [30], which involves most of the cell cycle biological activities [31]. Centromeric histone, Centromere Protein A (CENPA), a variant of canonical histone H3, plays an essential role in selective chromosome segregation in the cell cycle. Loading of CENPA protein at centromeres is closely associated with the cell cycle phases. When the cell proliferates, parental CENPA protein is deposited at centromeres in the S phase, whereas newly synthesized CENPA protein is deposited during the G2/M phase of the cell cycle [32–34]. A study reported that cell cycle-dependent deposition of CENPA was mediated by the Dos1/2–Cdc20 complex [35]. Although the cell cycle mechanisms involved in CENPA in cancer remain poorly studied, the function of CENPA in the cell cycle might be universal across all proliferating cells, regardless of their malignancy and tissue types, which inferred a potential common molecular pathological mechanism of CENPA shared across different cancer types.

Previous studies have reported the involvement of CENPA in a few cancer types. The over-expression of CENPA in prostate cancer has been demonstrated by a study with both in vivo and in vitro evidence [36]. In ovarian cancer, CENPA was found associated with the proliferation of cancer cells and survival of patients, which might be directly regulated by the MYBL2 [37]. In colonial cancer, CENPA was reported to recruit histone acetyltransferase general control of amino acid synthesis (GCN)-5 to the promoter region of the karyopherin α2 subunit gene (KPNA2), thereby boosting KPNα2 activation, which facilitated proliferation and glycolysis in cancer cells [38]. In clear cell renal cell carcinoma, the function of CENPA was reported to promote metastasis of cancer via the Wnt/β-catenin signaling pathway [39]. In addition, studies also suggested the prognostic value of CENPA for a few cancer types, such as ovarian cancer [37], liver cancer [40], breast cancer [41, 42], and lung cancer [43]. However, so far, a systematic pan-cancer bioinformatic analysis has not been done yet. Therefore, this study aimed to systematically investigate CENPA in multiple cancer types, regarding the potential of CENPA as a pan-cancer biomarker. Furthermore, we developed strategies for the application

of CENPA in glioma prognosis as an example of the future development of CENPA as a clinical cancer biomarker.

## Methods

### 1. The acquisition of mRNA sequencing data

The mRNA data, along with clinical information, were obtained from The TCGA [3], GTEx [4], CGGA [5], and the ICGC [6]. All data acquisition and usage adhered to the guidelines and policies of the respective databases. For TCGA, mRNA sequencing data across 33 cancer types were obtained via the TCGA portal. The CGGA data, which comprises three glioma patient cohorts, were also accessed through its portal. Corresponding normal tissue mRNA sequencing data for TCGA cancer types were downloaded from the GTEx portal.

### 2. Gene alteration analysis

Mutation analyses were performed using cBioPortal [44] with data from the "Pan-Cancer Analysis of Whole Genomes (ICGC/TCGA, Nature 2020)" [45]. Mutation and variant data were sourced from the TCGA PanCancer Atlas Studies and UniProt. Single-nucleotide variant (SNV) and copy number variant (CNV) data were retrieved from the NCI Genomic Data Commons (GDC) for TCGA datasets. SNV visualization was performed using the maftools package [46] which facilitated mutation frequency and variant type analysis. while CNV data were processed using GISTIC2.0 [47] to identify significant regions of amplification and deletion.

### 3. RNA-seq data analysis and plotting

All statistical analyses and visualizations were conducted using R version 4.0.3 (R Foundation for Statistical Computing, 2020). Nomogram construction, used to predict patient survival probabilities, was implemented with the rms package, which enabled the visualization of individualized risk scores. Kaplan-Meier (KM) survival analysis was performed to assess survival differences across groups, utilizing the survival package to generate survival curves and estimate hazard ratios with confidence intervals. Receiver Operating Characteristic (ROC) curves were constructed with the pROC package to evaluate the predictive accuracy of the biomarker, with area under the curve (AUC) values used as a measure of model performance. All plots, including survival curves and nomograms, were generated with ggplot2 (v3.3.2) for clear, publication-quality visualizations.

### 4. Associated genes enrichment analysis

The top correlated genes were identified using GEPIA [48], a tool that facilitates gene expression profiling and correlation analysis based on TCGA and GTEx data. A protein-protein interaction (PPI) network was then constructed using STRING [49], with a high-confidence interaction threshold (interaction score >0.9) to ensure robust connections between genes. Enrichment analyses, including Gene Ontology (GO) and Kyoto Encyclopedia of Genes and Genomes (KEGG) pathway analysis, were conducted using the clusterProfiler [50] package in R, which enabled the identification of significantly enriched biological processes, molecular functions, cellular components, and pathways associated with the gene set of interest.

### 5. Immunohistochemistry staining

Immunohistochemistry (IHC) staining was conducted using antibody CAB008371 on microarray slides to assess protein expression across cancerous and non-cancerous tissues.

Representative images were sourced from the Human Protein Atlas (HPA) [51], which provided high-quality, standardized IHC-stained samples from various tissue types on microarray slides. This setup allowed for a precise comparative analysis of protein expression, making it possible to observe differential expression patterns between cancerous and corresponding normal tissues, thereby facilitating insights into protein distribution and intensity across tissue types.

## 6. Immunofluorescence staining of cancer cells

Representative immunofluorescence staining images showing the subcellular distribution of the protein within the nucleus, endoplasmic reticulum (ER), and microtubules across three cancer cell lines were retrieved from the Human Protein Atlas (HPA) database [51],. These images illustrate the localization patterns of the protein within key cellular compartments, providing insights into its potential functional roles within the cell.

## 7. The cell cycle association analysis

The Human Protein Atlas (HPA) obtained and analyzed expression data plots from individual FUCCI U-2 osteosarcoma cells. Temporal mRNA expression patterns in these cells were characterized using the Fluorescent Ubiquitination-based Cell Cycle Indicator (FUCCI) U-2 OS cell line, which allows for precise tracking of cell cycle phases. This method enabled the observation of dynamic mRNA expression changes associated with distinct stages of the cell cycle, providing insights into cell cycle-dependent regulation of gene expression.

## 8. Stemness association analysis

The One-Class Logistic Regression (OCLR) algorithm [52] was employed to calculate the mRNA stemness index (mRNAsi) for TCGA pan-cancer mRNA sequencing data. This algorithm, specifically designed for single-class classification problems, works by learning a boundary that separates 'normal' data points (in this case, stemness-related features) from potential outliers or non-stemness signals in a high-dimensional space. By training on a set of stem cell-related genes, the OCLR algorithm identifies a boundary within the mRNA expression data that best characterizes stem-like properties across different cancer types. In the context of TCGA data, the OCLR model was trained on stem cell expression signatures and then applied to each tumor sample, assigning an mRNAsi score. This score reflects the degree of similarity between the tumor's gene expression profile and stem cell-like expression patterns, where higher mRNAsi values indicate stronger stemness characteristics. The mRNAsi thus serves as a quantitative measure of stemness, allowing for the comparison of stem-like properties across different cancer types and facilitating the exploration of how stemness contributes to tumor progression and heterogeneity.

## 9. Mutation association analysis

The mutation levels in the samples were assessed by calculating the Tumor Mutational Burden (TMB) [53] and evaluating Microsatellite Instability (MSI) status [54]. TMB, defined as the total number of mutations per megabase of a sequenced genome, was used as a quantitative measure of mutational load, providing insights into the genomic instability within each tumor sample. MSI status, an indicator of defects in DNA mismatch repair (MMR) mechanisms, was determined to identify tumors with high MSI, a characteristic often associated with increased mutation rates and potential immunogenicity. Together, TMB and MSI analyses enabled a

comprehensive assessment of mutation levels and mutational signatures across the cancer samples.

## 10. Immune cell infiltration analysis

Using the TCGA cohort, immune cell infiltration levels within tumor samples were estimated. This analysis was conducted using the CIBERSORT algorithm [55], a computational tool that quantifies the relative abundance of various immune cell types in complex tissues based on gene expression data. CIBERSORT deconvolutes bulk tumor transcriptomic data to infer the proportions of 22 distinct immune cell types, including T cells, B cells, macrophages, and natural killer cells. By applying this algorithm, we obtained a comprehensive profile of immune cell infiltration in each sample, allowing for further exploration of the tumor microenvironment's immune landscape and its potential association with clinical outcomes and cancer progression.

## 11. Single-cell sequencing data acquisition and analysis

Single-cell data were accessed and analyzed through the CancerSEA [56], CHARTS [57], and TISCH [58]. The datasets utilized included GSE117988 [59], GSE142213 [60], GSE143423, GSE131928 [61], GSE123814 [62], GSE70630 [63], etc.

## 12. Immune therapy prediction analysis

The Tumor Immune Dysfunction and Exclusion (TIDE) algorithm was used to perform immune therapy prediction analysis [64]. TIDE is a computational framework that evaluates the potential for immune evasion by simulating dysfunction in T cells and exclusion mechanisms within the tumor microenvironment. To assess the biomarker relevance of CENPA compared to standardized cancer immune evasion biomarkers, we examined its expression across immune checkpoint blockade (ICB) sub-cohorts and visualized these comparisons in a bar plot. The predictive performance of CENPA and other biomarkers regarding ICB response status was further evaluated by calculating the area under the receiver operating characteristic curve (AUC), which provided a quantitative measure of their accuracy in distinguishing responders from non-responders to ICB therapies.

## 13. Drug screening and prediction

Drug screening was conducted by evaluating the correlation between CENPA expression and drug sensitivity, applying a stringent significance cutoff of $p < 1e\text{-}5$. The area under the dose-response curve (AUC) values, reflecting drug efficacy, were analyzed alongside CENPA expression profiles across various cancer cell lines using GSCALite [65]. For this analysis, drug sensitivity data from the Genomics of Drug Sensitivity in Cancer (GDSC) [66] and Cancer Therapeutics Response Portal (CTRP) [67] databases were integrated, providing a comprehensive dataset for evaluating the sensitivity of different drugs in relation to CENPA expression. Spearman correlation analysis was then applied to determine the association between the expression levels of genes in the selected gene set and the sensitivity of small molecules/drugs.

To support the investigation of drug interactions with CENPA, a predictive structural model of the CENPA protein was retrieved from the AlphaFold database [68]. Protein-ligand docking was conducted using AutoDock Vina (version 1.1.2) [69], employing cavity-detection-guided blind docking to identify potential binding sites within the protein structure. This approach enabled the prediction of interaction sites and binding affinities, providing insights into potential therapeutic targets involving CENPA.

### 14. Statistical analysis

Gene expression differences were compared using either the Wilcox test or the Kruskal-Wallis test. Survival analysis was performed using Kaplan-Meier analysis, along with the log-rank test and Cox regression test. Pearson's correlation test was applied to assess the relationship between two variables. Statistical significance was determined with a threshold of $P<0.05$.

## Results

### 1. Genomic alteration of CENPA in cancers

The initial analysis of this study focused on investigating CENPA genomic alterations in various cancers. The alteration frequency bar plot revealed that the total alteration frequencies in most cancer types were below 10%. Non-small cell lung cancer exhibited the highest frequency, at 15.2% (7 out of 46 cases). The majority of gene alterations were amplifications (S1A Fig). To further explore CENPA mutations in cancers, TCGA mutation data was plotted, indicating that CENPA harbored only a low number of single-nucleotide variants across cancers (S1B Fig), which is consistent with the previous findings. The analysis of copy number variation demonstrated that nearly all copy number alterations of CENPA were heterozygous. Most cancer types exhibited 20–40% of CENPA heterozygous amplification, and approximately half had 5–10% heterozygous deletion samples. Lung squamous cell carcinoma (LUSC) showed the highest percentage of CENPA heterozygous amplification with no instances of heterozygous deletion, aligning with the earlier observation of a high frequency of gene amplification in lung cancer. In contrast, kidney chromophobe (KICH) had about 60% heterozygous deletions of CENPA with no amplification (S1C Fig). Overall pan-cancer data indicated that CENPA copy number could influence mRNA expression (S1D Fig). Therefore, while CENPA gene mutations may not be the primary driver in most cancers, its copy number alterations might influence cancer development through changes in mRNA expression.

### 2. The overexpression of CENPA in cancers

The analysis demonstrated that CENPA was overexpressed in the majority of cancer types compared to normal tissues in both females and males (Fig 1A). To streamline data presentation, abbreviations were used to denote cancer types, which are listed in S1 Table. The mRNA expression analysis of CENPA, utilizing data from TCGA and GTEx, revealed significant overexpression in 30 out of the 33 cancer types examined. Notably, mesothelioma (MESO) and uveal melanoma (UVM) lacked comparable normal tissue, while acute myeloid leukemia (LAML) was the only cancer type where CENPA expression was lower in cancerous tissues than in normal tissues (Fig 1B). To achieve better control in the comparison between cancerous and non-cancerous tissues, paired samples from the same patients were analyzed. This comparison indicated that CENPA was significantly overexpressed in 16 cancer types (Fig 1C). To further investigate CENPA overexpression in cancers, protein staining of CENPA in cancerous versus corresponding normal tissues was examined in representative cancer types. The staining images generally showed that, although CENPA staining intensity was slightly stronger in cancer tissues, the overall staining intensity in both cancerous and normal tissues was low, potentially due to the properties of the antibody used (Fig 1D).

### 3. The diagnostic value of CENPA in cancers

To assess the diagnostic value of CENPA in various cancers, single-variable receiver operating characteristic (ROC) curves were plotted for different cancer types, and the area under the curves (AUC) was calculated using data from TCGA and GTEx. The results demonstrated that

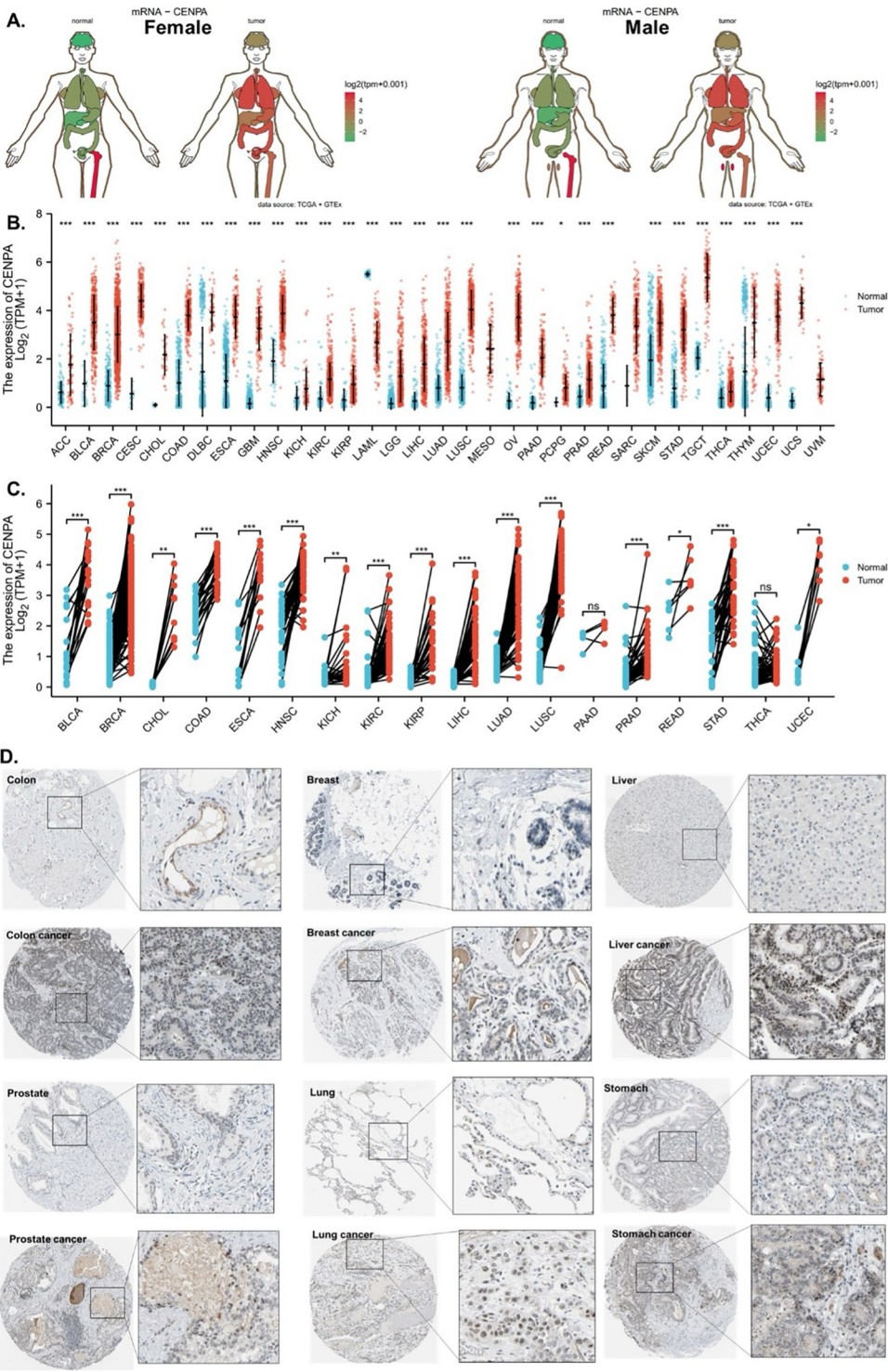

**Fig 1. The overexpression of CENPA in cancers. A.** Anatomy plot of the gene expression profile of CENPA across all tumor samples and paired normal tissues in females and males. TCGA data were plotted. **B.** The gene expression profile of CENPA across all tumor samples and normal tissues. TCGA and GTEx data were plotted. **C.** Paired sample expression profile of CENPA across all tumor samples and normal tissues. TCGA data were plotted. **D.** Representative protein staining images of CENPA in cancers and corresponding normal tissues. The images were downloaded from the Human Protein Atlas (HPA). *p<0.05; **p<0.01; ***p<0.001.

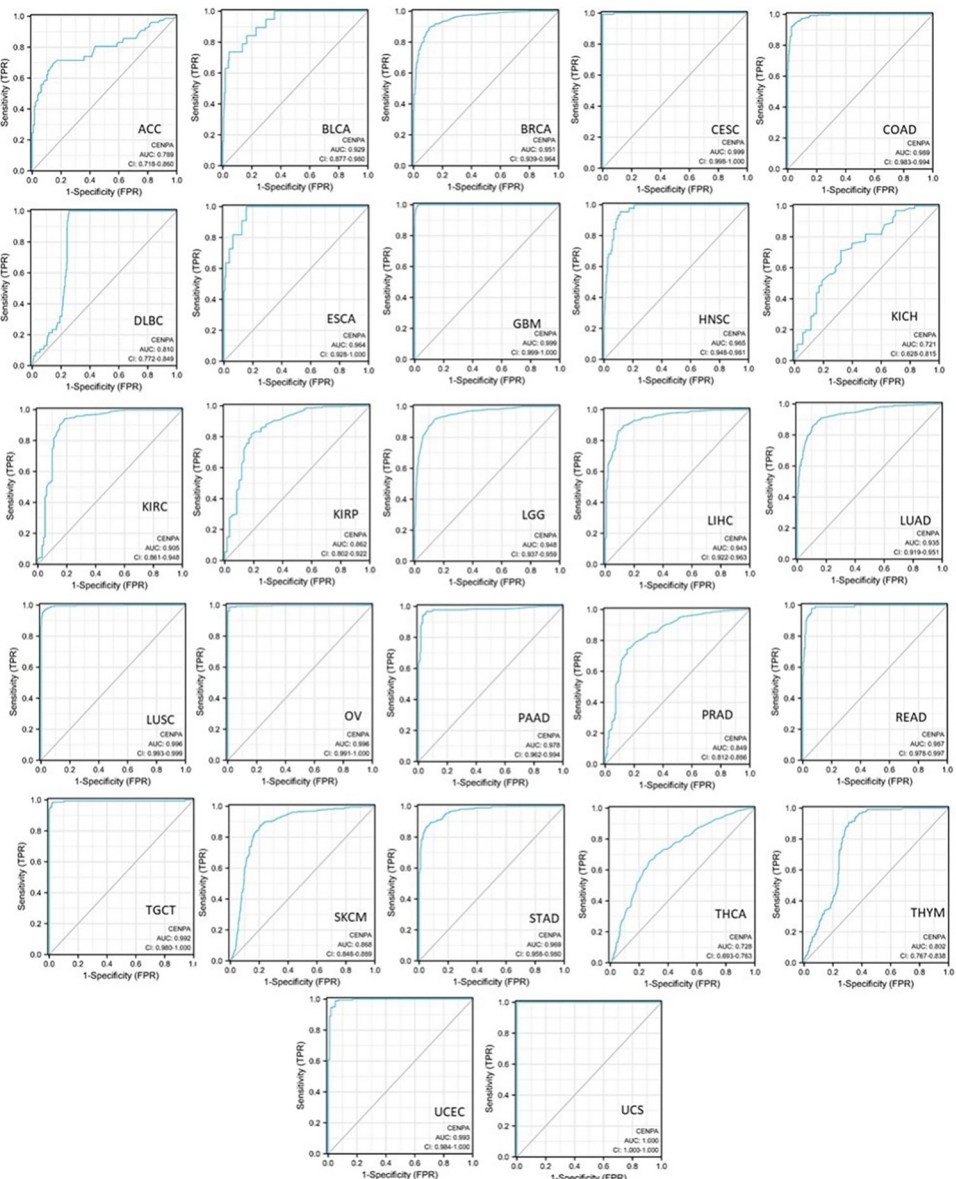

**Fig 2. The pan-cancer diagnostic value of CENPA.** The diagnostic receiver operating characteristic (ROC) curve of different cancer types. TCGA and GTEx data were used to calculate the ROC. The area under the curves (AUC) and the corresponding 95% confidential interval (95%CI) was shown.

19 cancer types had AUCs exceeding 0.9, indicating an outstanding diagnostic power of CENPA. Five cancer types had AUCs ranging from 0.8 to 0.9, supporting the excellent diagnostic capability of CENPA. Additionally, three cancer types had AUCs between 0.7 and 0.8, reflecting an acceptable diagnostic power of CENPA [70] (Fig 2). These results suggested that CENPA is a promising diagnostic molecular biomarker that can be developed for multiple cancer types.

## 4. The prognostic value of CENPA in cancers

This study also aimed to explore the prognostic value of CENPA in various cancers. To this end, univariate overall survival Cox regression analysis was performed for CENPA across 33

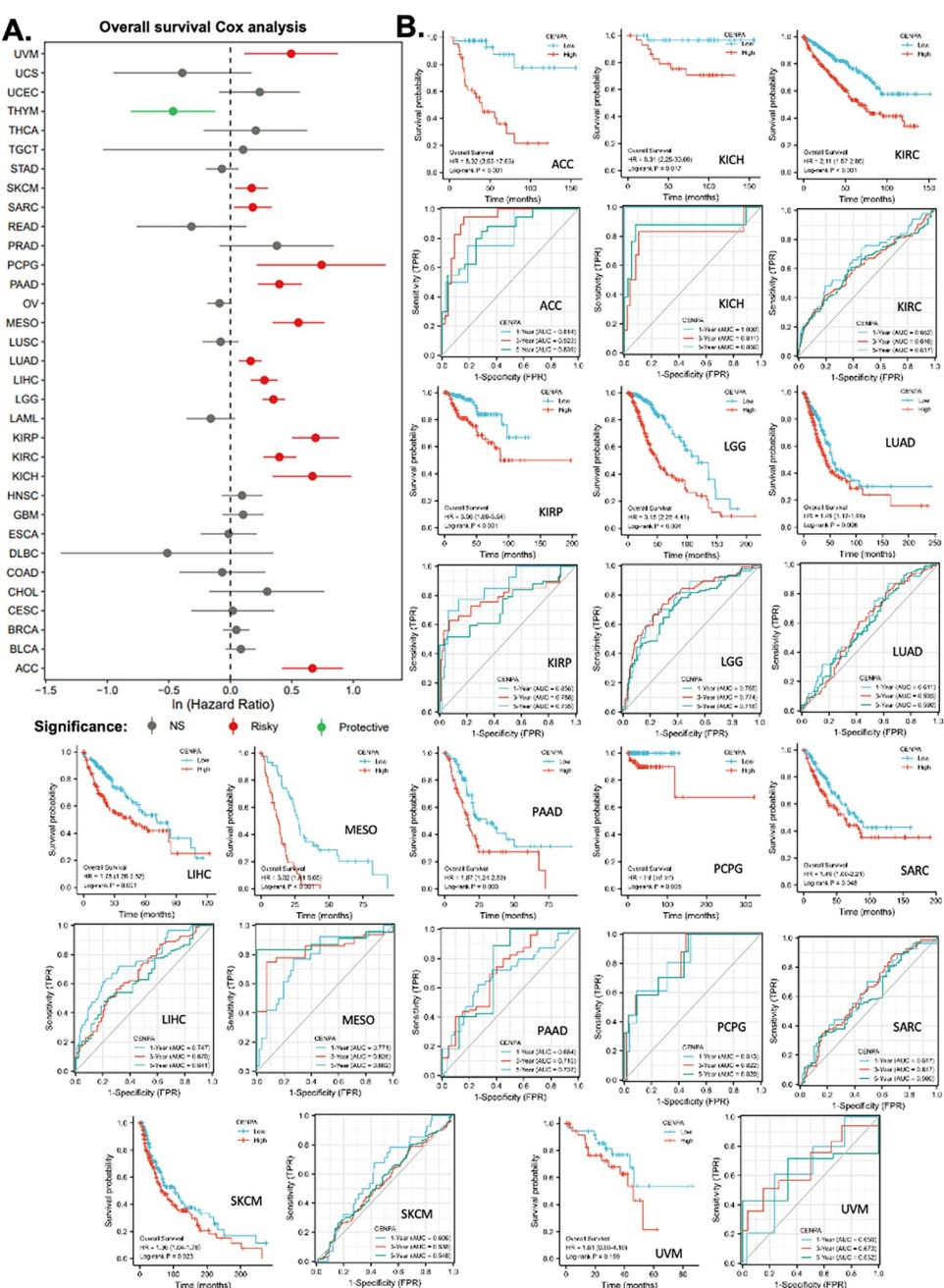

**Fig 3. The pan-cancer prognostic value of CENPA.** TCGA data were analyzed. **A.** Univariate Cox regression analysis of CENPA for overall survival in different cancer types. **B.** The overall survival Kaplan-Meier (KM) plot and log-rank analysis of high (50–100%) and low (0–50%) CENPA patients with time-dependent (1-, 3-, and 5-year) overall survival prognostic receiver operating characteristic curve (ROC). Only cancer types with significance in Cox regression were plotted.

cancer types using TCGA data. The results revealed that CENPA was significantly associated with worse overall survival in 13 cancer types, while it was linked to better overall survival in one cancer type, thymoma (THYM) (Fig 3A). To further investigate the association between CENPA and overall survival, Kaplan-Meier (KM) plots and log-rank analyses were conducted for the cancer types that showed significance in the Cox regression analysis. The results

indicated that 12 cancer types remained significant in the log-rank analysis (Fig 3B, first panel for each cancer type).

To assess the prognostic value of CENPA in these cancer types, time-dependent prognostic ROC curves were plotted. For 1-year overall survival, the AUC for kidney chromophobe (KICH) exceeded 0.9, indicating outstanding predictive power. The AUCs for adrenocortical carcinoma (ACC), kidney renal papillary cell carcinoma (KIRP), and pheochromocytoma and paraganglioma (PCPG) ranged between 0.8 and 0.9, suggesting excellent predictions. The AUCs for lower-grade glioma (LGG), liver hepatocellular carcinoma (LIHC), and mesothelioma (MESO) were between 0.7 and 0.8, indicating acceptable predictions. For 3-year overall survival, the AUC for adrenocortical carcinoma (ACC) was over 0.9, indicating outstanding predictive accuracy. The AUCs for kidney chromophobe (KICH), mesothelioma (MESO), and pheochromocytoma and paraganglioma (PCPG) ranged between 0.8 and 0.9, suggesting excellent predictions, while the AUCs for kidney renal papillary cell carcinoma (KIRP), lower-grade glioma (LGG), and pancreatic adenocarcinoma (PAAD) were between 0.7 and 0.8, indicating acceptable predictions. For 5-year overall survival, the AUCs for adrenocortical carcinoma (ACC), kidney chromophobe (KICH), mesothelioma (MESO), and pheochromocytoma and paraganglioma (PCPG) were between 0.8 and 0.9, indicating excellent predictive power, while the AUCs for kidney renal papillary cell carcinoma (KIRP), lower-grade glioma (LGG), and pancreatic adenocarcinoma (PAAD) were between 0.7 and 0.8, suggesting acceptable predictions (Fig 5B, second panel for each cancer type). These findings suggest that CENPA is a promising prognostic molecular biomarker with potential applicability in multiple cancer types, such as adrenocortical carcinoma (ACC), kidney chromophobe (KICH), kidney renal papillary cell carcinoma (KIRP), lower-grade glioma (LGG), mesothelioma (MESO), and pheochromocytoma and paraganglioma (PCPG).

## 5. The application of CENPA for glioma prognosis

To demonstrate the practicable clinical application of CENPA, we focused on one cancer type, glioma, where CENPA was demonstrated to have promising prognostic value. The World Health Organization (WHO) defined glioma into four grades based on histology and clinical criteria: G1, G2, G3, and G4 [71]. The G1 glioma is generally benign and has a very good prognosis, which has been distinguished from the G2, G3, and G4 glioma. In TCGA cohort, G2 and G3 glioma together are referred to as "low-grade glioma (LGG)", while G4 glioma is referred to as "glioblastoma multiforme (GBM)" (highest grade glioma) [72]. In this context, this study combined LGG and GBM and analyzed the prognostic value of CENPA for overall glioma.

To validate the prognostic accuracy of CENPA for overall survival in glioma patients, we examined its prognostic association across five independent glioma datasets: TCGA (LGG +GBM) (n = 703), CGGA mRNAseq693 (n = 693), CGGA mRNAseq325 (n = 325), CGGA mRNA-array301 (n = 301), and ICGC (pediatric brain tumor) (n = 120). Kaplan-Meier (KM) plots and Cox regression analyses demonstrated that high CENPA expression was significantly associated with worse survival across all five datasets. The hazard ratios (HR) ranged from 2.95 to 7.21. ROC analysis revealed that for 1-year overall survival prediction, four datasets indicated acceptable accuracy. For 3-year overall survival prediction, three datasets suggested excellent accuracy, while two indicated acceptable accuracy. For 5-year survival prediction, three datasets showed excellent accuracy, and two indicated acceptable accuracy (Fig 4A).

In this study, we developed strategies for applying CENPA in glioma prognosis, illustrating its potential as a clinical prognostic biomarker for cancer. To identify variables for the CENPA-based prognostic model in glioma patients, we performed Cox regression analysis to

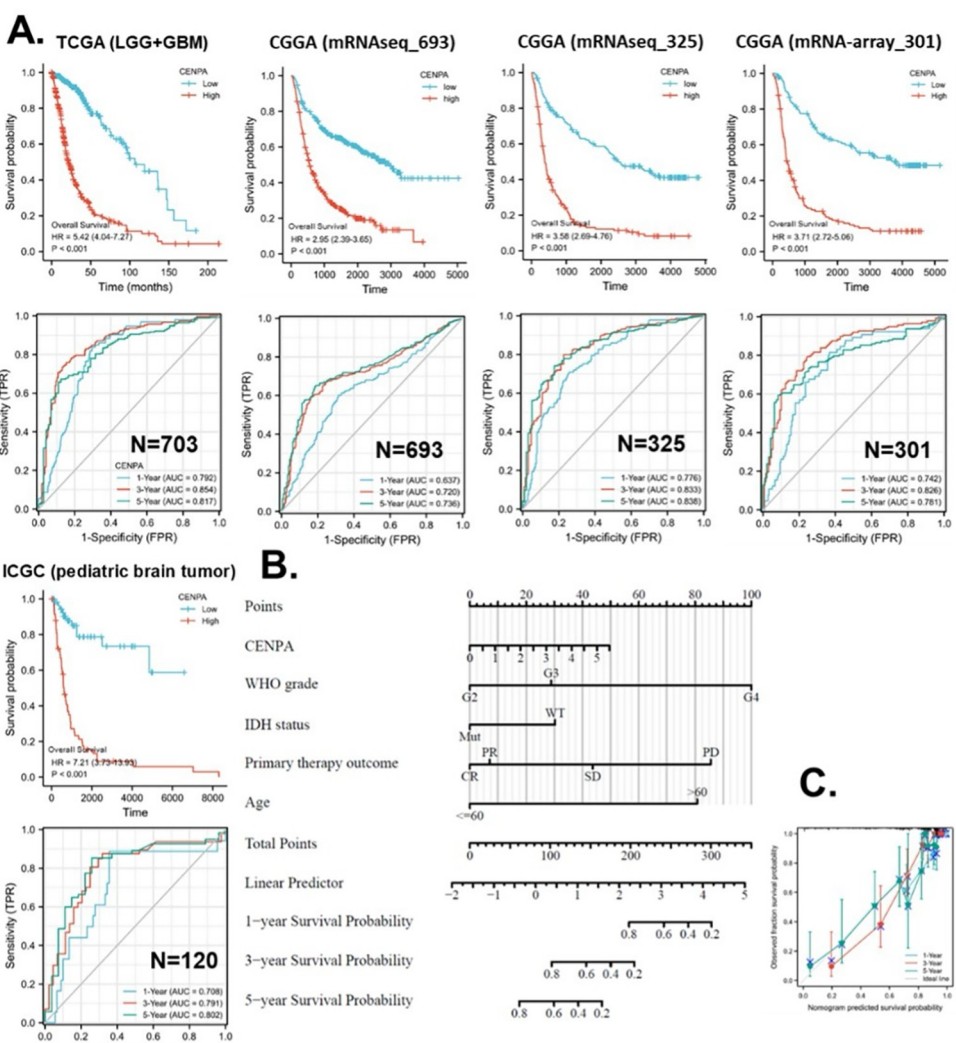

**Fig 4. Application of CENPA for glioma prognosis. A.** validation of the survival association of CENPA in five independent glioma cohorts. TCGA (LGG+GBM), CGGA (mRNAseq 693), CGGA (mRNAseq 325), CGGA (mRNA-array 301), and ICGC (pediatric brain tumor) were analyzed. The overall survival Kaplan-Meier (KM) plot and Cox analysis of high (50–100%) and low (0–50%) CENPA patients with time-dependent (1-, 3-, and 5-year) overall survival prognostic receiver operating characteristic curve (ROC) were shown. **B.** Nomogram for the prediction of 1-, 3-, and 5-year overall survival of glioma patients. The TCGA (LGG+GBM) cohort was used to construct the prognostic model of CENPA for glioma. **C.** Calibration plots of the nomogram for estimation of overall survival of glioma patients at years 1, 3, and 5.

evaluate prognostic factors. Univariate Cox regression results indicated that CENPA level, 1p/19q codeletion, primary therapy outcome, IDH status, and age were significantly associated with overall survival in glioma patients. Multivariate Cox regression showed that CENPA level, primary therapy outcome, IDH status, and age remained significant after adjustment for other variables, suggesting they provide additional prognostic power as independent factors in the model (S2 Table). Consequently, these factors, along with WHO grade (G2-4), were included in the prognostic model for overall survival in glioma patients. Based on this model, a nomogram was constructed to predict the survival probability of glioma patients at 1, 3, and 5 years (Fig 4B). The calibration curves of the nomogram predictions generally aligned with the observed outcomes in patients (Fig 4C).

## 6. CENPA was highly expressed in the nucleus of malignant cells

To explore the cell populations and cellular locations where CENPA is expressed, we conducted an analysis of single-cell sequencing data and observed the subcellular distribution of CENPA through immunofluorescence staining in three cancer cell lines. The single-cell sequencing data set included three cancer types: acute erythroid leukemia (AEL), breast cancer (BRCE), glioma, and Merkel cell carcinoma (MCC). The analysis revealed that CENPA was expressed by a small subset of malignant cells, whereas immune cells exhibited relatively low levels of CENPA expression (Fig 5A). Immunofluorescence staining of the subcellular distribution of CENPA in prostate cancer cell line PC-3, rhabdomyosarcoma cell line RH30, and osteosarcoma cell line U2OS demonstrated that CENPA was predominantly localized in the nucleus, although U2OS exhibited relatively lower fluorescence intensity (Fig 5B). It is worth noting that rhabdomyosarcoma is a type of sarcoma. Prostate cancer (PRAD) and sarcoma (SARC) were shown earlier in this study to overexpress CENPA, while osteosarcoma was not included among the cancer types in the TCGA data set.

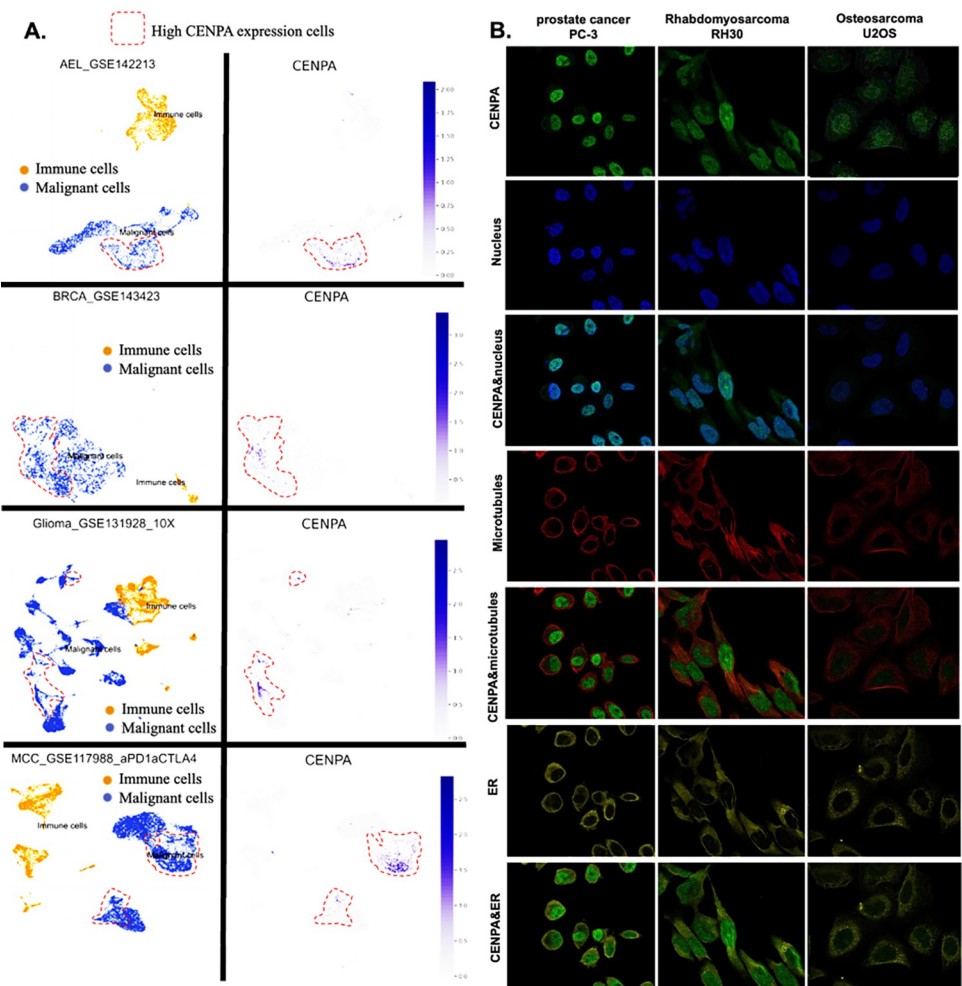

**Fig 5. The expression of CENPA in cell populations and cellular locations. A.** The expression of CENPA in cell populations in cancer tissues. Single-cell mRNA expression cohorts were accessed and analyzed using the TISCH. **B.** Immunofluorescence staining of the subcellular distribution of CENPA within the nucleus, endoplasmic reticulum (ER), and microtubules of three cancer cell lines.

## 7. CENPA was associated with the cell cycle of cancer cells

Since CENPA was predominantly detected in the nucleus of cancer cells, we hypothesized two potential roles for CENPA in cancers: 1) CENPA may influence the mutation of other genes, given that gene transcription occurs in the nucleus, and 2) CENPA could regulate the cell cycle, as DNA replication during the cell cycle also takes place in the nucleus. To test the first hypothesis, we analyzed the correlation between CENPA expression and two mutation indicators: tumor mutation burden (TMB) and microsatellite instability (MSI). TMB quantifies the approximate number of gene mutations within the cancer genome, while MSI reflects a state of genetic hypermutability resulting from impaired DNA mismatch repair (MMR). The presence of MSI serves as phenotypic evidence that MMR is not functioning correctly. The analysis indicated that CENPA expression was positively correlated with TMB and MSI across most cancer types, though the correlations were weakly significant (S2A, S2B Fig). These findings suggest that CENPA is not generally associated with genomic instability in cancers.

To explore the potential common functional effects of CENPA in cancers, we identified the top CENPA-correlated genes by analyzing data from all 33 TCGA cancer types as a single cohort. The top 30 CENPA-correlated genes were used to construct a protein-protein interaction (PPI) network, highlighting the possible associations between CENPA and these genes (S2C Fig). Further analysis of the top 200 correlated genes was conducted through GO and KEGG enrichment studies. KEGG pathway enrichment revealed that the top two pathways associated with CENPA were "DNA replication" and "Cell cycle." The top GO molecular function (MF) was related to ATPase activities, the top GO cellular component (CC) was chromosome regions, and the top GO biological processes (BP) included "organelle fusion," "mitotic nucleus division," and "nucleus division." These GO-enriched terms were all linked to cancer proliferation and the cell cycle (S2D Fig).

To further validate the potential association between CENPA and cancer proliferation and cell cycle regulation, we analyzed the correlation between CENPA expression and cancer functional signals using multiple single-cell data sets across various cancer types. These correlation results were summarized (as shown in the top bar plot of S2E Fig) to provide an overview of CENPA's potential common roles in cancers. The results indicated that the most significant positive correlations were with "cell cycle" and "proliferation," supporting the hypothesis that CENPA may regulate the cell cycle and proliferation. Additionally, CENPA appeared to be negatively associated with "apoptosis," "DNA repair," and "metastasis" (S2E Fig). These data support the notion that CENPA may play a role in regulating cancer growth.

## 8. CENPA is a biomarker for the cell cycle G2 phase in cancer cells

The ability of a tumor to proliferate and propagate relies on a small population of stem-like cells, the OCLR algorithm [52] has been wildly applied for the estimation of the stemness in a tissue sample. In this study, the mRNAsi (a measure of stemness) was calculated for 33 cancer types in the TCGA, and the correlation between CENPA expression and pan-cancer stemness was analyzed. The results indicated that CENPA expression was positively correlated with stemness across most cancer types (Fig 6A), suggesting that the association with stemness might be a common mechanism of CENPA in cancer. Building on these findings, we proposed that CENPA could serve as a novel cell cycle biomarker and conducted a GSEA enrichment analysis of CENPA-correlated genes in the "REACTOME CELL CYCLE CHECKPOINTS" pathway. As expected, the analysis showed that CENPA-correlated genes were significantly enriched in "REACTOME CELL CYCLE CHECKPOINTS" (Fig 6B).

To further understand CENPA's specific role in different phases of the cell cycle in cancer cells, we analyzed single-cell expression data for CENPA across various cell cycle phases in

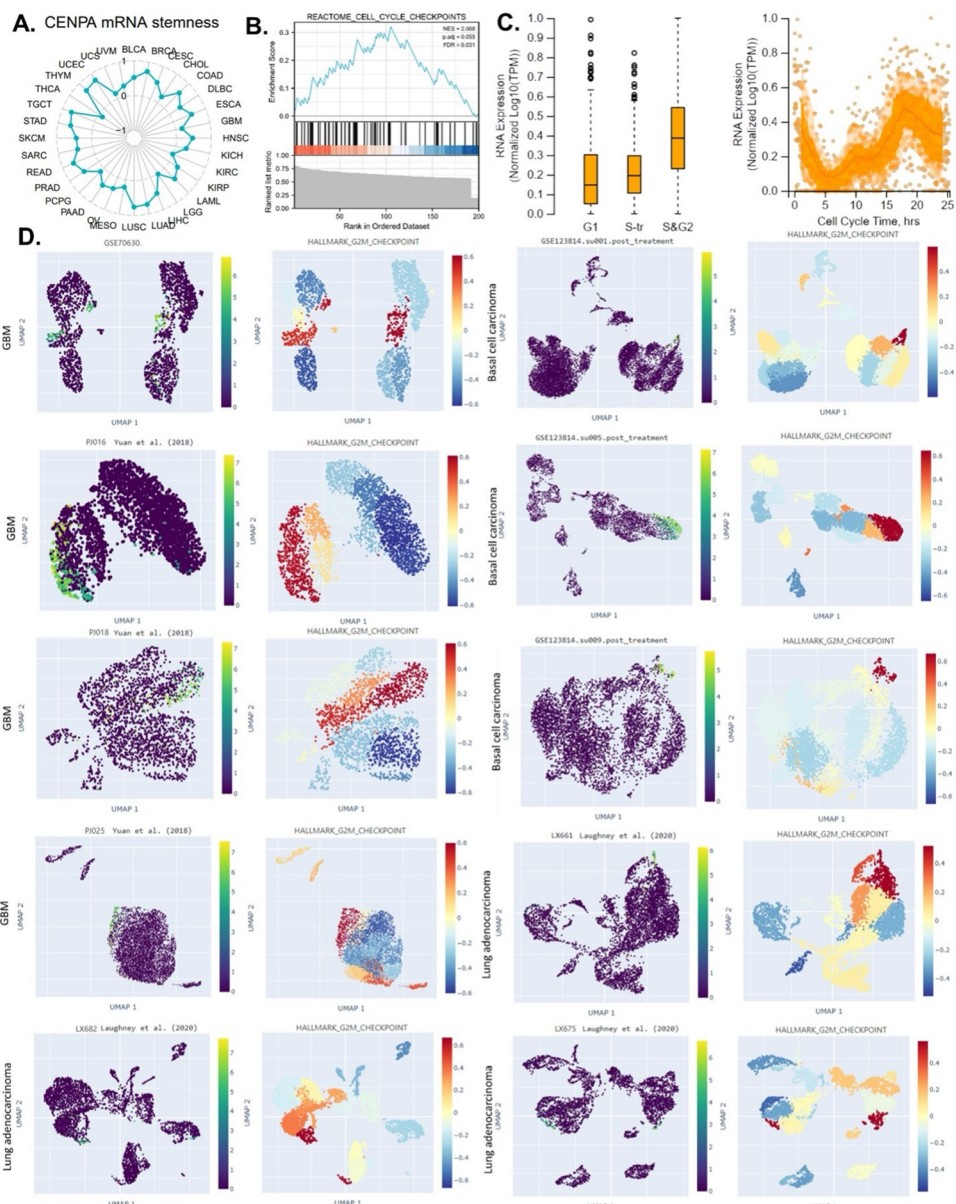

**Fig 6. The potential of CENPA as a cell-cycle biomarker for the M/G2 phase in cancers. A.** The correlation of OCLR scores and CENPA in TCGA cancer data. The OCLR algorithm was used to calculate the mRNAsi (OCLR scores) for the evaluation of stemness. **B.** The GSEA enrichment of CENPA-correlated genes in "REACTOME CELL CYCLE CHECKPOINTS". The top 200 CENPA-correlated genes were identified using the GEPIA based on all TCGA cancer data and used for the GSEA enrichment analysis. **C.** Plots of single-cell RNA-sequencing data from the FUCCI U-2 OS osteosarcoma cell line, showing the correlation between CENPA mRNA expression and cell cycle progression. **D.** The expression of CENPA in single cells and the G2M checkpoint hallmark signals in cancer tissues. Single-cell data were accessed and analyzed using the CHARTS.

U2OS cells, which predominantly express CENPA in the nucleus. The results revealed that CENPA expression was low during the G1 phase and high during the S and G2 phases (Fig 6C). Based on these findings, we hypothesized that CENPA might be closely associated with the G2 phase of the cell cycle. To test this hypothesis, we examined CENPA expression across several single-cell cancer datasets and compared it with single-cell signals of the G2/M

checkpoint, a hallmark gene set related to cell proliferation in GSEA [73]. Among all the ten single-cell cancer data sets analyzed, CENPA was highly expressed in a population of cell clusters that had strong signals of G2M checkpoint. These results confirmed that CENPA was a biomarker for the cell cycle G2 phase (Fig 6D).

## 9. The immune microenvironment association of CENPA in cancers

This study also investigated the potential of CENPA as a biomarker for the immune microenvironment. Since the effectiveness of cancer immune therapy largely depends on immune cell infiltration levels and the presence of immune checkpoints, we explored the value of CENPA as a predictive biomarker for immune therapy from these two perspectives.

Earlier analyses revealed that CENPA was predominantly expressed in a small population of malignant cells, with relatively low expression in immune cells. However, whether CENPA expression in tumors affects immune cells has not been previously examined. To address this, we calculated immune cell infiltration levels in cancers and analyzed their correlation with CENPA expression. The analysis identified T cell CD4+ as the most notable immune cell type correlated with CENPA; Th2 cells were positively correlated with CENPA across all cancer types, and Th1 cells were positively correlated in the majority of cancers. Additionally, common lymphoid progenitors showed a positive correlation with CENPA in most cancer types. CENPA was closely associated with multiple immune cells across different cancers, particularly in lung squamous cell carcinoma (LUSC), lung adenocarcinoma (LUAD), glioblastoma multiforme (GBM), and thymoma (THYM) (Fig 7A).

We also examined the correlation between CENPA and several commonly used immune checkpoints in current immune therapies. The results indicated that CENPA was positively associated with most immune checkpoints in thyroid carcinoma (THCA), lung adenocarcinoma (LUAD), liver hepatocellular carcinoma (LIHC), lower-grade glioma (LGG), kidney renal clear cell carcinoma (KIRC), breast invasive carcinoma (BRCA), and bladder urothelial carcinoma (BLCA), while it showed a negative correlation with most immune checkpoints in thymoma (THYM), lung squamous cell carcinoma (LUSC), glioblastoma multiforme (GBM), cervical squamous cell carcinoma and endocervical adenocarcinoma (CESC), and adrenocortical carcinoma (ACC) (Fig 7B).

To compare the predictive performance of CENPA for immune checkpoint blockade (ICB) treatment with other standardized biomarkers, we assessed the relevance of CENPA and other biomarkers based on their ability to predict ICB response outcomes in various sub-cohorts. The results showed that CENPA expression had an AUC greater than 0.5 in 11 out of 25 ICB sub-cohorts, which is higher than the number of cohorts where microsatellite instability (MSI) score, tumor mutational burden (TMB), T cell clonality (T.Clonality), and B cell clonality (B. Clonality) achieved an AUC over 0.5 (seven, nine, and six cohorts, respectively). However, the predictive value of CENPA was lower than that of CD27A, tumor immune dysfunction and exclusion (TIDE), interferon-gamma (IFNG), CD8, and Merck 18 (Fig 7C). These comparisons highlight the potential value of CENPA as a predictive biomarker for immune therapy.

## 10. Computational drug predictions of CENPA in cancers

Given that our study demonstrated a close association between CENPA and the cancer cell cycle, patient survival, and the immune microenvironment, we proposed CENPA as a potential therapeutic target for cancer drug treatment. To explore this, we screened and predicted potential drugs targeting CENPA using cancer drug databases and computational methods. Drug sensitivity data were obtained from the GDSC and CTRP databases, and the correlation between CENPA expression and the sensitivity of cancer cell lines to various small molecules

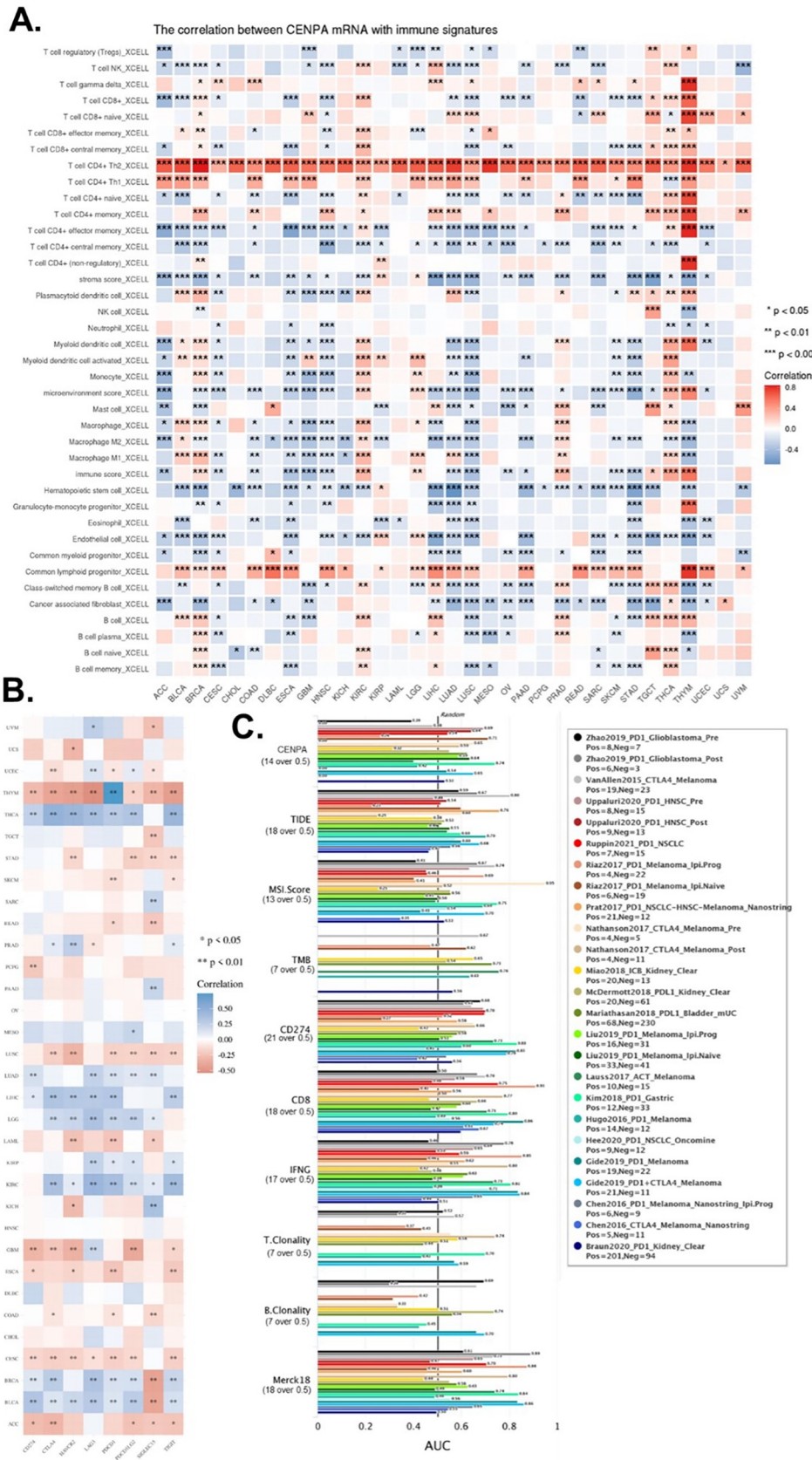

**Fig 7. The immune microenvironment association of CENPA in cancers. A.** The correlation of CENPA expression and immune cell infiltration levels. TCGA data were analyzed. The Xcell algorithms were used to estimate the immune cell infiltration levels. **B.** The correlation of CENPA expression and immune checkpoint genes expression. TCGA data were analyzed. **C.** Bar plot showing the biomarker relevance of CENPA compared to standardized cancer immune evasion biomarkers in immune checkpoint blockade (ICB) sub-cohorts. The area under the receiver operating characteristic curve (AUC) was applied to evaluate the predictive performances of the biomarkers on the ICB response status.

and drugs was analyzed. Data from multiple cancer cell lines in GDSC and CTRP were integrated for these calculations. We applied a significance cutoff of p<1e-5 to identify relevant drugs. The screening identified 8 drugs with sensitivities negatively correlated with CENPA levels in cancer cells and 4 drugs with sensitivities positively correlated with CENPA levels (Fig 8A and S3 Table). We hypothesized that these 12 drugs might directly interact with the CENPA protein.

To predict the direct interaction between CENPA and these 12 drugs, we accessed the predictive protein structural model of CENPA from the Alphafold database and performed protein-ligand docking for CENPA and the identified drugs. The predicted aligned error of the CENPA protein structure model indicated that the N-terminus had a long tail with low model confidence, while the docking was focused on regions with very high model confidence (Fig 8B, 8C). A protein-ligand model with a vina score lower than -8 was considered to have a very good binding affinity. The docking results revealed that CD-437, 3-Cl-AHPC, Trametinib, BI-2536, and GSK461364 had high binding affinities to CENPA (S3 Table), suggesting that these drugs are likely to directly target CENPA in cancer cells. All docking models are displayed in Fig 8D.

## Discussion

This study used bioinformatic data to support the potential values of CENPA for clinical cancer diagnosis and prognosis. Although the function of CENPA in cell growth and cell cycle has been studied [74], the association of CENPA and cancers has not been studied comprehensively and the clinical use of CENPA as a biomarker for cancer has not been developed. CENPA has been proposed as a genomic marker for centromere activity [75]. Single-cell analysis in this study suggested that CENPA was highly expressed during the S&G2 phase in the cell cycle and was closely associated with the G2/M checkpoint in cancer cells. These indicated that CENPA can be a biomarker for the G2 phase in the cell cycle.

In addition, CENPA plays a central role in the regulation of centromere activity. The inheritance of genetic material requires the faithful segregation of chromosomes during cell division, when kinetochores, a unique centromere macromolecular protein, attach chromosomes to the spindle for proper movement and segregation. CENPA directly regulates the assembly of active kinetochores, thereby regulating cell division [76]. While this process is crucial for nearly all proliferating cells, irrespective of whether they are malignant, one common characteristic of cancer cells is their significantly higher proliferation rate compared to non-cancerous cells. This suggests that cancer cells undergo more frequent cell divisions and, therefore, might require higher levels of CENPA for proper kinetochore regulation. The expression analysis supported this hypothesis, showing that cancer cells indeed express higher levels of CENPA compared to non-cancerous cells. The results revealed that almost all cancer types overexpressed CENPA, with the exception of LAML, which exhibited lower CENPA expression in cancerous tissues than in normal tissues. This finding is understandable, as LAML, being a type of leukemia, likely has distinct cell cycle regulation mechanisms compared to most other cancer types [77]. The overexpression of CENPA in cancers inferred its pan-cancer

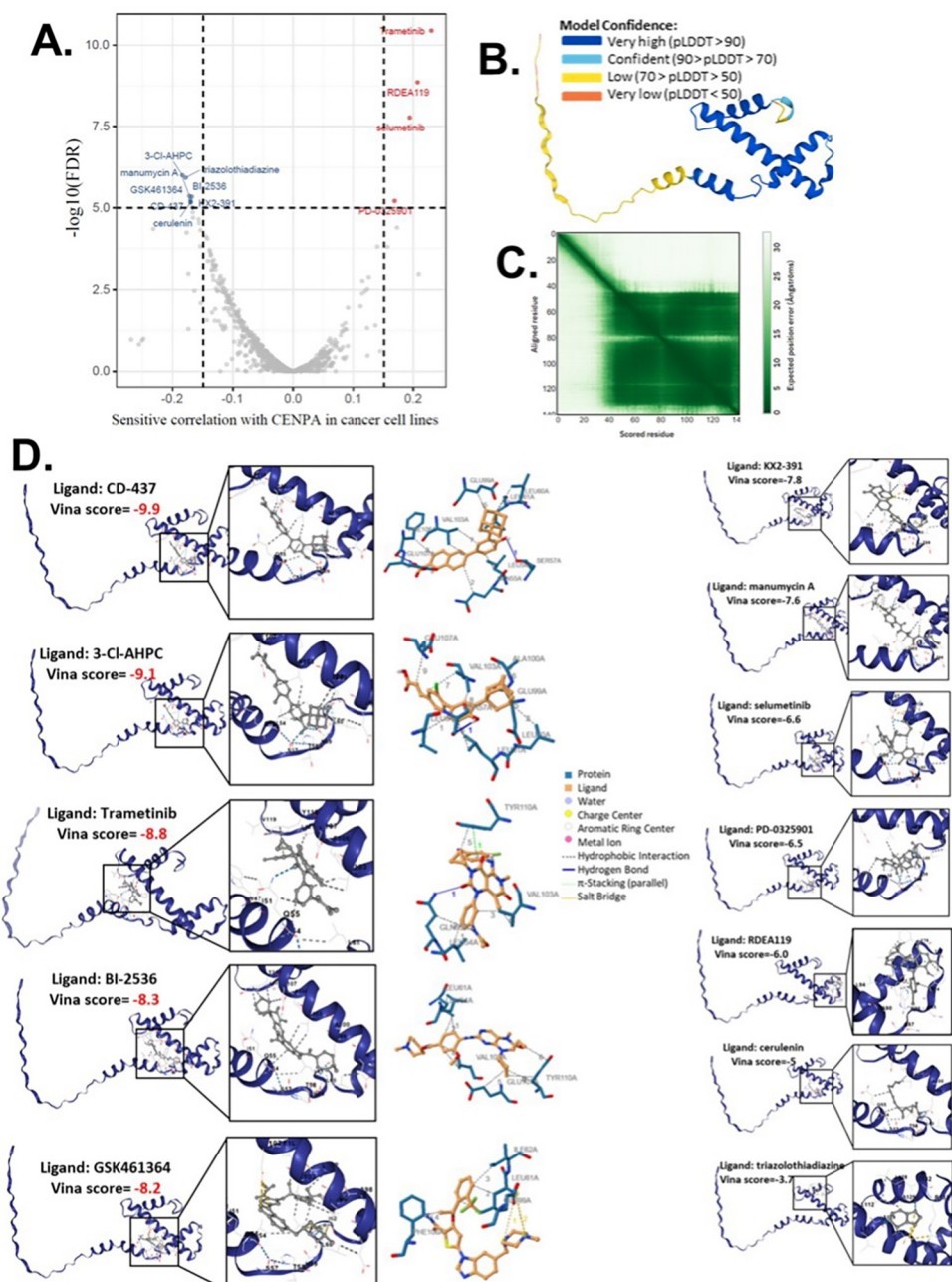

**Fig 8. The computational drug prediction of CENPA in cancers. A.** The volcano plot of the correlation of CENPA expression and small molecule/drug sensitivity of cancer cell lines. GDSC and CTRP data were analyzed. Drug sensitivity and gene expression profiling data of multiple cancer cell lines in GDSC and CTRP were integrated for investigation. The expression of CENPA was performed by Spearman correlation analysis with the small molecule/drug sensitivity (area under the IC50 curve). **B.** Predictive protein structural model of CENPA. **C.** Predicted aligned error of the CENPA protein structure model. **D.** Protein-ligand docking models of CENPA and identified drugs. The names of the ligands and the docking vina scores were shown. For models with a vina score of lower than -8.0 (indicates a binding affinity), the protein-ligand molecular interaction profiles were displayed on the right.

potential as a diagnostic biomarker and a therapeutic target. Nevertheless, further studies to compare the diagnostic power of CENPA with present diagnostic biomarkers are required for further development of CENPA for clinical use.

The gene alteration analysis in this study indicated that CENPA mutations are unlikely to be a major driving factor in cancer development, given the low mutation rate observed. However, changes in copy number could potentially influence cancer progression by increasing the transcription of CENPA mRNA. As a result, our study primarily focused on the expression levels of CENPA. A previous study has reported that overexpression of CENPA can promote genome instability in human cells, particularly when the retinoblastoma protein is inactivated [78]. Our TMB and MSI analysis indicated that CENPA was not associated with genome instability in all cancers. In eye cancer (UVM), CENPA was not correlated with TMB but correlated with MSI. The analysis of single-cell data (S2E Fig) also suggested that CENPA was negatively correlated with DNA repair in eye cancer. Most of these results were consistent with the previous study.

The expression of CENPA has been reported to be associated with worse overall survival of some cancer types, such as ovarian cancer [37], liver cancer [40], breast cancer [41, 42], and lung cancer [43]. Most of these studies were also using TCGA data, but they were only limited to one cancer type regardless of the common roles of CENPA across multiple cancer types. A previous study had demonstrated the potential of CENPA as a prognostic biomarker for GBM [79]. However, the conclusions of the previous study were based solely on TCGA data and focused exclusively on GBM. In contrast, this study extends those conclusions to glioma as a whole, encompassing both low-grade and high-grade gliomas. The prognostic association between CENPA and glioma patient survival was supported by five independent glioma datasets, with sample sizes of 703, 693, 325, 301, and 120, respectively. Given the relatively larger number of datasets and independent data sources, we believe that the prognostic performance of CENPA is quite reliable.

The immune association of CENPA in certain cancer types, such as lung and liver cancer [80], has been previously demonstrated using TCGA data [43]. This study broadened the scope of this association to a pan-cancer context and compared CENPA's predictive value for immune checkpoint blockade (ICB) response with that of other immune response biomarkers. While the ICB cohorts in this study were not large, the findings suggest a potential role for CENPA in predicting immune therapy outcomes, which warrants further validation. Additionally, we used computational methods to screen and predict potential drugs targeting CENPA in cancer cells. These computational predictions also require experimental validation to confirm their efficacy.

## Conclusion

CENPA holds promise as a biomarker in cancers linked to cell cycle regulation and stemness, with significant potential for diagnostic, prognostic, and therapeutic applications.

## Supporting information

**S1 Table. List of the cancer type abbreviations.**
(DOCX)

**S2 Table. Cox regression analysis of CENPA and clinical characteristics in glioma.**
(DOCX)

**S3 Table. Computational drug predictions of CENPA.**
(DOCX)

**S1 Fig. Alteration of CENPA in cancers genome.** TCGA cohort was analyzed.
(DOCX)

**S2 Fig. Functional and mutation associations of CENPA in cancers.**
(DOCX)

## Author Contributions

**Conceptualization:** Panpan Wang.

**Data curation:** Tao Tang.

**Formal analysis:** Miray Karsidag.

**Funding acquisition:** Miray Karsidag.

**Methodology:** Hengrui Liu.

**Resources:** Kunwer Chhatwal.

**Software:** Hengrui Liu.

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
