## [Decision Letter · Decision Letter 0]

29 Oct 2024

PONE-D-24-37517Single-cell and bulk RNA sequencing analysis reveals CENPA as a biomarker and therapeutic target in cancersPLOS ONE

Dear Dr. Liu,

Thank you for submitting your manuscript to PLOS ONE, which has been assessed by two independent reviewers. After careful consideration, we feel that it has merit but does not fully meet PLOS ONE’s publication criteria as it currently stands. Therefore, we invite you to submit a revised version of the manuscript that addresses the points raised during the review process.

We look forward to receiving your revised manuscript.

Kind regards,

Zhiming Li, Ph.D.

Academic Editor

PLOS ONE

Journal Requirements:

3. Please note that your Data Availability Statement is currently missing the repository name and/or the DOI/accession number of each dataset OR a direct link to access each database. If your manuscript is accepted for publication, you will be asked to provide these details on a very short timeline. We therefore suggest that you provide this information now, though we will not hold up the peer review process if you are unable.

Reviewers' comments:

Reviewer's Responses to Questions

**Comments to the Author**

1. Is the manuscript technically sound, and do the data support the conclusions?

Reviewer #1: Yes

Reviewer #2: No

2. Has the statistical analysis been performed appropriately and rigorously? 

Reviewer #1: Yes

Reviewer #2: I Don't Know

3. Have the authors made all data underlying the findings in their manuscript fully available?

Reviewer #1: Yes

Reviewer #2: Yes

4. Is the manuscript presented in an intelligible fashion and written in standard English?

Reviewer #1: Yes

Reviewer #2: Yes

5. Review Comments to the Author

Reviewer #1: The manuscript presents a comprehensive bioinformatic analysis of CENPA expression across various cancers, evaluating its potential as a biomarker for cancer diagnosis and prognosis. The authors have conducted a thorough investigation, examining not only CENPA's roles in cancer but also its correlation with the tumor microenvironment. Additionally, they have performed screening to identify potential drugs targeting CENPA in cancer cells.

The study design is logical and well-structured, with comprehensive and convincing data analysis that supports the conclusions drawn.

To further enhance the manuscript, I suggest the following revisions:

1. In sections 2, 7, and 9, please write out the full names of cancer types rather than using abbreviations. This would improve readability for a broader audience.

2. The cell type labels in Figure 2A are currently too small to read clearly. Please increase the font size to ensure legibility.

3. The resolution of Figure 5B needs improvement, as the current image appears slightly blurry. A higher-resolution version would better showcase the results.

Overall, these minor revisions would enhance the presentation of this otherwise well-executed study.

Reviewer #2: Here, a pan-cancer bioinformatic analysis has been performed on centromere protein A (CENPA). The manuscript analyzed different aspects in different cancer types without a logical procedure. The methods used are not well described, the results are confusing and do not support the conclusions.

6. PLOS authors have the option to publish the peer review history of their article (what does this mean?). If published, this will include your full peer review and any attached files.

Reviewer #1: **Yes: **Yin Tang

Reviewer #2: No

---

## [Author Response · Author response to Decision Letter 0]

3 Nov 2024

Reviewer #1: The manuscript presents a comprehensive bioinformatic analysis of CENPA expression across various cancers, evaluating its potential as a biomarker for cancer diagnosis and prognosis. The authors have conducted a thorough investigation, examining not only CENPA's roles in cancer but also its correlation with the tumor microenvironment. Additionally, they have performed screening to identify potential drugs targeting CENPA in cancer cells.

The study design is logical and well-structured, with comprehensive and convincing data analysis that supports the conclusions drawn.

To further enhance the manuscript, I suggest the following revisions:

1. In sections 2, 7, and 9, please write out the full names of cancer types rather than using abbreviations. This would improve readability for a broader audience.

We have written out the full names of cancer types rather than just using abbreviations.

2. The cell type labels in Figure 2A are currently too small to read clearly. Please increase the font size to ensure legibility.

We have increased the size of the cell type labels.

3. The resolution of Figure 5B needs improvement, as the current image appears slightly blurry. A higher-resolution version would better showcase the results.

We have increased the resolution of Figure 5B.

Overall, these minor revisions would enhance the presentation of this otherwise well-executed study.

Reviewer #2: Here, a pan-cancer bioinformatic analysis has been performed on centromere protein A (CENPA). The manuscript analyzed different aspects in different cancer types without a logical procedure. 

We appreciate your comment and understand your concern about the logical procedure. Therefore, we have rearranged the order of presenting the results, now the paper follows logic below:

1 comparison between cancer non-cancer to support the diagnostic value

2 analysis of survival association for supporting the prognostic value 

3 glioma as a representative cancer typo to support the practical clinical application of CENPA for prognosis.

4 explore the mechanism underlying the impact of CENPA on cancer,

1) first, we find it is expressed in malignant cells not immune cells

2) it is expressed in nucleur

3) it is highly associated with cancer cell stemness (proliferation)

5 explore the how the gene impact the immune cells and exam if it is useful for prediction of immune therapy.

6 drug prediction of CENPA in cancers

The methods used are not well described, 

We have modified the method description and included much more details. If it is still need to be improve, please specify and we are happy to modify it.

the results are confusing and do not support the conclusions.

We have rearranged the results and modified the conclusion to 

“CENPA holds promise as a biomarker in cancers linked to cell cycle regulation and stemness, with significant potential for diagnostic, prognostic, and therapeutic applications.”

We also modified the title and included “potential” to avoid overstate of the conclusion.

---

## [Decision Letter · Decision Letter 1]

18 Nov 2024

Single-cell and bulk RNA sequencing analysis reveals CENPA as a potential biomarker and therapeutic target in cancers

PONE-D-24-37517R1

Dear Dr. Liu,

We’re pleased to inform you that your manuscript has been judged scientifically suitable for publication and will be formally accepted for publication once it meets all outstanding technical requirements.

Kind regards,

Zhiming Li, Ph.D.

Academic Editor

PLOS ONE

Additional Editor Comments (optional):

Reviewers' comments:

Reviewer's Responses to Questions

**Comments to the Author**

1. If the authors have adequately addressed your comments raised in a previous round of review and you feel that this manuscript is now acceptable for publication, you may indicate that here to bypass the “Comments to the Author” section, enter your conflict of interest statement in the “Confidential to Editor” section, and submit your "Accept" recommendation.

Reviewer #1: All comments have been addressed

Reviewer #2: All comments have been addressed

2. Is the manuscript technically sound, and do the data support the conclusions?

Reviewer #1: Yes

Reviewer #2: Yes

3. Has the statistical analysis been performed appropriately and rigorously? 

Reviewer #1: Yes

Reviewer #2: Yes

4. Have the authors made all data underlying the findings in their manuscript fully available?

Reviewer #1: Yes

Reviewer #2: Yes

5. Is the manuscript presented in an intelligible fashion and written in standard English?

Reviewer #1: Yes

Reviewer #2: Yes

6. Review Comments to the Author

Reviewer #1: Thanks for addressing all previous comments and concerns raised during the initial review. The changes have significantly improved the clarity and quality of the manuscript.

Reviewer #2: The authors rearranged the manuscript by better describing the methods used and following a comprehensive logical presentation. The results are clear and support the conclusions.

7. PLOS authors have the option to publish the peer review history of their article (what does this mean?). If published, this will include your full peer review and any attached files.

Reviewer #1: **Yes: **Yin Tang

Reviewer #2: No

---

## [Editor Report · Acceptance letter]

8 Dec 2024

PONE-D-24-37517R1 

PLOS ONE

Dear Dr. Liu, 

I'm pleased to inform you that your manuscript has been deemed suitable for publication in PLOS ONE. Congratulations! Your manuscript is now being handed over to our production team.

Kind regards, 

on behalf of

Dr. Zhiming Li 

Academic Editor

PLOS ONE